# Ocular Drug Delivery: A Special Focus on the Thermosensitive Approach

**DOI:** 10.3390/nano9060884

**Published:** 2019-06-14

**Authors:** Simona Sapino, Daniela Chirio, Elena Peira, Elena Abellán Rubio, Valentina Brunella, Sushilkumar A. Jadhav, Giulia Chindamo, Marina Gallarate

**Affiliations:** 1Department of Drug Science and Technology, University of Turin, 10125 Turin, Italy; elena.peira@unito.it (E.P.); giulia.chindamo@unito.it (G.C.); marina.gallarate@unito.it (M.G.); 2NIS Research Centre, University of Turin, 10125 Turin, Italy; valentina.brunella@unito.it (V.B.); sushil.unige@gmail.com (S.A.J.); 3Jesús Usón Minimally Invasive Surgery Centre, 10071 Cáceres, Spain; eabellan@ccmijesususon.com; 4Department of Chemistry, University of Turin, 10125 Turin, Italy; 5School of Nanoscience and Technology, Shivaji University Kolhapur, Maharashtra 416004, India

**Keywords:** posterior segment diseases, ocular drug delivery, hydrogels, nanocomposites, thermosensitive systems, in vitro pharmacokinetic models

## Abstract

The bioavailability of ophthalmic therapeutics is reduced because of the presence of physiological barriers whose primary function is to hinder the entry of exogenous agents, therefore also decreasing the bioavailability of locally administered drugs. Consequently, repeated ocular administrations are required. Hence, the development of drug delivery systems that ensure suitable drug concentration for prolonged times in different ocular tissues is certainly of great importance. This objective can be partially achieved using thermosensitive drug delivery systems that, owing to their ability of changing their state in response to temperature variations, from room to body temperature, may increase drug bioavailability. In the case of topical instillation, in situ forming gels increase pre-corneal drug residence time as a consequence of their enhanced adhesion to the corneal surface. Otherwise, in the case of intraocular and periocular, i.e., subconjunctival, retrobulbar, peribulbar administration, among others, they have the undoubted advantage of being easily injectable and, owing to their sudden thickening at body temperature, have the ability to form an in situ drug reservoir. As a result, the frequency of administration can be reduced, also favoring the patient’s adhesion to therapy. In the main section of this review, we discuss some of the most common treatment options for ocular diseases, with a special focus on posterior segment treatments, and summarize the most recent improvement deriving from thermosensitive drug delivery strategies. Aside from this, an additional section describes the most widespread in vitro models employed to evaluate the functionality of novel ophthalmic drug delivery systems.

## 1. Introduction

One of the critical issues of topically administered ophthalmic drugs is that their efficacy is limited by the fast drug clearance due to pre-corneal fluid drainage; consequently, frequent administrations are required. Therefore, various drug delivery systems (DDS) have received increased attention to enhance the efficacy of drugs on the corneal surface.

Aside from this, many limitations make it hard to deliver drugs aimed to treat eye posterior segment diseases, such as diabetic retinopathy and age-related macular degeneration (AMD). In fact, topical ocular medications do not reach the back of the eye; moreover, systemic administration is rarely used because of the small volume of the eye and the presence of the blood retinal barrier (BRB) [1].

A lot of research is currently being done to improve transscleral delivery, which might offer the advantage of removing the need to breach the walls of the eye; many transscleral delivery systems, also associated to iontophoresis, are therefore at different stages of development. However, to date, the majority of treatments of the posterior segment, such as retinal and choroidal disorders, require the intravitreal pathway. Intravitreal injection (IVI) is currently considered the most validated option—although it is invasive and associated with serious side effects—for the delivery of large molecules such as anti-vascular endothelial growth factors (anti-VEGF antibodies), whose use has reached an exponential growth in recent years due to the progressive expansion of their clinical applications [2]. Nevertheless, the periocular pathway, including the retrobulbar, peribulbar, subtenon and subconjunctival routes, and even topical delivery, continue to be explored.

To overcome the limitations of conventional eye drops in corneal/conjunctival administration, and of invasive injection in intraocular administrations, or of surgery-implanted cannulas in periocular administration, in the last decades, several ophthalmic formulations, such as drug-loaded hydrogels and contact lenses targeted to the anterior segment, or ocular implants and physical devices destined to the back of the eye, have been proposed.

Hydrogels are three-dimensional, cross-linked networks of either synthetic or natural water-soluble polymers with great potential in several applications, such as drug delivery, cell encapsulation and tissue engineering [3]. Various advantages make them interesting—their aqueous environment can exert some protection towards cells and labile drugs (such as peptides, proteins, DNA and oligonucleotides) and they have a significant role in transporting nutrients to cells. As a result, they are attractive for various ophthalmic applications, among which are corrective soft contact lenses [4], adhesives for ocular wound repair [5], potential vitreous substitutes [6] and drug vehicles [7]. Regarding the latter application, to our knowledge, most hydrogels on the market are targeted to the anterior segment (such as Pilopine HS, Zirgan and Pilogel), due to their ability to increase viscosity and mucoadhesive properties [8]. Conversely, just few hydrogels have been already approved by the FDA and EMA for intraocular injectable applications; as an example, Akten (Akorn, Buffalo Grove, IIlinois), a 3.5% lidocaine gel, was approved by the FDA for all ophthalmic procedures in October 2008, including intraocular procedures [9,10].

Other hydrogels are currently on the market for ophthalmic application other than drug delivery purposes—as an example, ReSure Sealant is an in situ gel approved to seal clear corneal incisions following cataract surgery [11].

Recently, contact lenses for drug controlled delivery to the anterior chamber have been developed, but several challenges are still arising regarding the limited drug release, the strict regulatory issues and the high cost of clinical studies [12].

On the other hand, the crucial need to reach the eye posterior segment through less invasive strategies other than repeated injections has boosted the development of slow-release implants that can be placed at once at various ocular sites. Currently, intraocular implants that allow sustained drug release in the posterior segment are at different development stages [13]. Most of these consist of non-biodegradable polymers, such as silicone, polyvinyl alcohol and ethylene vinyl acetate, from which long-lasting release of the entrapped drugs occurs [14]. However, they require surgical intervention and need removal or replacement by new implants. On the contrary, biodegradable polymers such as poly(lactic) acid and poly(lactic-*co*-glycolic) acid offer the advantage of releasing the drug at the same time that the polymer degrades in the target site, avoiding the need of surgical removal [15,16].

To date, several ocular implants designed for the treatment of severe indications affecting the posterior segment of the eye, including macular degeneration, are on the market or are undergoing clinical trials. Among them, non-biodegradable Vitrasert, Retisert, Medidur, Iluvien and biodegradable Posurdex, Ozurdex and Surodex must be cited [17,18,19,20]. Most of these implants are loaded with small active compounds, such as fluocinolone acetonide, dexamethasone and ganciclovir; meanwhile no biologics-carrying implants are available in the market, some being, however, in the pipeline [21].

Anti-VEGF therapy, playing a central role in the pathogenesis of choroidal neovascularization, has revolutionized the medical management of diabetic retinopathy and of AMD [22]. Currently, the most common anti-VEGF agents are pegaptanib (Macugen), bevacizumab (Avastin) and ranibizumab (Lucentis) [23,24], followed by other emerging macromolecular drugs already in clinical trials, among these being Fovista (Ophthotech, Princeton, NJ, USA), a platelet-derived growth factor aptamer currently in phase III clinical trials [25], and designed ankyrin repeat proteins [26].

The use of intravitreal administration in anti-VEGF therapy is still presenting some problems—most drugs are rapidly cleared from the vitreous humor, inducing the need of repeated injections that can cause side effects, such as endophthalmitis, retinal detachment, hemorrhage and poor patient tolerance [27]; other drugs induce local toxicity when administered at their effective dose, causing side effects and possible retinal lesions [28]. For these reasons, strategies that can deliver sufficient drug concentrations to this anatomic region in a less invasive manner and with less frequent doses, such as sustained-release DDS, represent an area of active interest in the ophthalmology community. In the last decades, different technologies have been proposed to this aim, including the use of nanomedicine [29,30,31]. Therapies based on nanotechnologies, such as lipid and polymeric nanocarriers, present several advantages, allowing a precisely targeted drug delivery and controllable release of the therapeutics [32]; moreover, the stability and half-life in the vitreous of entrapped drugs might be enhanced, thus reducing the frequency of administration and, consequently, diminishing their toxicity [33]. Therefore, depending on particle charge, surface properties and relative hydrophobicity, nanoparticles (NP) can be designed to be successfully used in sustained ocular therapy [34], both in the anterior and posterior segments [35]. Studies have shown that albumin NP can serve as a very efficient drug delivery system for retinal diseases, such as cytomegalovirus retinitis, as they are biodegradable, non-toxic and have non-antigenic properties [36]. Moreover, NP prepared with natural polymers, such as chitosan, increased the intraocular penetration of loaded drugs, due to their ability to make contact intimately with corneal and conjunctival surfaces [37]. In the past decades, several hydrophilic polymeric particles have been proposed as ocular DDS composed of various biodegradable polymers, such as poly(lactic acid) [38], poly(alkyl cyanoacrylate) (PACA) [39], poly(lactic-*co*-glycolic acid) (PLGA) [40] and poly(ɛ-caprolactone) (PECL) [41]. However, one of the main barrier-hindering clinical trials of these innovative systems is the requirement to ensure the safety of nano-microsystems and of their biodegradation products in the eye [29,42].

Another appealing approach of drug delivery to the posterior ocular segment consists in vesicular systems such as intravitreal-injectable liposomes (i.e., the ocular liposomal Verteporfin (Visudyne). They provide sustained drug delivery for weeks or even months, but up until today, most of them are only pre-clinically investigated, with few in clinical use [43].

Recently, preclinical trials have centered around the interesting formulation of nanocomposites, consisting of nanoparticulate systems dispersed into a hydrogel matrix that provide an additional diffusion barrier to drug release, eliminating the burst effect and extending the release profiles of the entrapped drugs [44].

In the literature, various strategies have been proposed to deliver drugs into the eye in a more controlled manner—in the first part of this review, special attention will be given to the thermosensitive approach, considering the different typologies and action mechanisms. The second part deals with the most widespread in vitro models employed to investigate the functionality of novel ophthalmic DDS.

## 2. Thermosensitive DDS

In recent years, significant progress has been made to control and target drug delivery by designing “smart” systems capable of releasing a loaded drug as a response to a specific stimulus. When these DDS are administered, drug release is activated by some local input (changes in pH level, electric and magnetic fields, light, temperature, etc.). The stimuli responsible for activating drug release can differ according to the nature or the type of applied energy—temperature is certainly the most exploited one for ophthalmic application, and injectable thermoresponsive polymers have especially been proposed. Being characterized by the existence of a critical temperature, these systems undergo reversible transition (due to volume collapse or phase transition to a solid or gel state) in the physiological temperature range. Thereby, it would be possible to introduce them into the body with minimal invasion by circumventing the need for a surgical procedure for placement of the system. Thus, thermosensitive systems represent a very promising approach that can have practical applications in ophthalmic research leading the advances in controlled drug delivery [28].

The following paragraphs give an overview of the most significant formulations (Table 1) able to form an in situ drug reservoir responding to room-to-body temperature change, and of their relevant applications in ophthalmic drug delivery, focusing upon the posterior segment.

### 2.1. In Situ Thermosensitive Hydrogels

In situ thermosensitive hydrogels are generally based on natural polymers, such as polysaccharides (cellulose and chitosan) or synthetic polymers (including PEG, poly(*N*-isopropylacrylamide) (PNIPAM) and Pluronic F-127) crosslinked by a variety of mechanisms [45]. Before administration they are aqueous solutions gelling after a temperature change, owing to a temperature-induced phase transition governed by the balance of hydrophilic and hydrophobic moieties. These systems, exhibiting low viscosity at 23 °C and forming gels at 37 °C, have been exploited in the ophthalmic field, both for anterior and posterior segment treatments.

One of the major drawbacks in the treatment of anterior segment diseases, commonly managed with topical medications, is the difficulty to provide and to maintain an adequate drug concentration in the precorneal area, as a consequence of the physiological eye removal mechanisms (lacrimation and blinking). Thus, there is the need of an ideal dosage form able to prolong its retention time on the eye surface.

Among the various vehicles proposed by the literature, thermo-gelling formulations are gaining increasing interest, owing to the obvious advantages presented by shifting to the gel phase once instilled. Several in situ gelling systems have been developed to prolong drug precorneal residence time [46]. Sandri et al. [47] developed thermosensitive eye drops, based on chondroitin sulphate associated with hydroxypropylmethyl cellulose, containing platelet lysate for the treatment of corneal ulcers. Cho et al. prepared a brimonidine-loaded thermosensitive formulation of hexanoyl glycol chitosan for glaucoma therapy that exhibited attractive bioavailability of the drug, as well as a more prolonged period of lowered intra-ocular pressure [48]. More recently, a thermo-sensitive in situ gelling formulation of ketoconazole based on poly(*N*-isopropylacrylamide)/hyaluronic acid was prepared and tested on New Zealand white rabbits for in vivo antimicrobial studies—the increased residence time was confirmed and a controlled drug release was obtained following topical administration in the eye [49].

As reported above, the anatomy and safeguard barrier of the cornea compromise the rapid absorption of drugs, thus making the topical treatment of the posterior segment quite impractical, and making intraocular injections only just preferable that, as reported above, unfortunately can cause heavy side effects. In situ injectable thermogelling systems have been extensively investigated with the aim of reducing the frequency of intraocular injection acting as sustained drug release depot [50,51]. Various polymers are capable of forming thermosensitive injectable hydrogels with a transition temperature of ∼32 °C. PNIPAM and its copolymers are among the most intensively investigated for ophthalmic application, primarily for their biocompatibility [52,53]. Derwent and colleagues employed PNIPAM cross-linked with poly(ethylene glycol) diacrylate (PEG-DA) to successfully encapsulate proteins, including bovine serum albumin (BSA), immunoglobulin G (IgG) and, finally, bevacizumab and ranibizumab. These kinds of hydrogels exhibited fast and reversible phase changes with alteration in temperature, and their release profiles resulted as a function of the cross-link density [54,55]. In other research studies, copolymers based on acrylic acid *N*-hydroxysuccinimide (NAS), *N*-isopropylacrylamide (NIPAM) and increasing concentrations of acryloyloxy dimethyl-γ-butyrolactone (DBA) and acrylic acid (AA) were synthesized with the aim to obtain thermoresponsive, resorbable copolymers suitable to deliver drugs and/or cells to the posterior segment of the eye in a minimally invasive manner [56,57]. Copolymers of NIPAM and NAS were also conjugated with cell adhesive peptides and amine-functionalized hyaluronic acid. These novel copolymers were designed by Mazumder et al. [58] to provide non-invasive delivery of therapeutic cells into the hard-to-access sub-retinal space for the treatment of retinal diseases. More recently, bevacizumab was delivered in a biodegradable, thermoresponsive hydrogel, poly(ethylene glycol)-poly-(serinol hexamethylene) ESHU, that, according to in vitro/in vivo experiments, resulted well-tolerated and capable of sustaining bevacizumab release [59,60].

PEG-PLA-based hydrogels have also been considered, and many attempts have been taken to find those with a critical gelation temperature (LCGT) resembling body temperature [61]. PLGA-PEG-PLGA copolymer was found to be an efficacious thermosensitive in situ gel-forming material for ophthalmic drug delivery [62]. In vivo pharmacokinetic studies showed that, through an intravitreal injection of PLGA-PEG-PLGA hydrogel in adult male Sprague Dawley rats, the overtime release of bevacizumab in the vitreous humor and retina was greatly extended [63]. The same (PLGA–PEG-PLGA) triblock copolymer was used as an injectable biocompatible matrix for the intravitreal release of hydrophobic glucocorticoid dexamethasone (DEX). It was possible to conveniently modulate its in vitro release rate from the thermogel by varying the mixing ratio of the two copolymers [64]. Interestingly, Patel and colleagues successfully synthesized an injectable in situ hydrogel composed of biodegradable pentablock copolymers, such as PEG, PCL and PLA for controlled ocular protein delivery. Compared to triblock copolymers, pentablock copolymers exhibit significantly reduced crystallinity and higher biocompatibility [65].

The poly(ethylene oxide)-poly(propylene oxide)-poly(ethylene oxide) (PEO-PPO-PEO) triblock copolymers, known as Pluronics (BASF) or Poloxamers (ICI), have been widely used as non-ionic surfactants, solubilizers and DDS. Because of the lower critical gelation concentration and the lowest toxicity in the commercially available Pluronics series, Pluronic F127 (Poloxamer 407) has been studied most extensively as a DDS. Nonetheless, Pluronic hydrogels have several drawbacks, such as poor gel durability, weak mechanical strength and rapid drug release. To overcome such disadvantages, Pluronic-based grafted polymers have been proposed as an injectable and sustained delivery system of human growth hormone [66]. Among these, an amine-terminated Pluronic F127 was grafted with hyaluronic acid (HA). The obtained Pluronic-HA hydrogel showed a lower critical gelation concentration and dissolution rate, with a consequently slow drug release rate [67]. Bhoyar and colleagues evaluated different gelling solutions made with Poloxamers in combination with several polymers (HPMC, Chitosan, PVP, PVA) for the sustained intravitreal release of ciprofloxacin. Increase in Poloxamer concentration within the range of 18–24% *w*/*w* showed a decrease in phase transition temperature. The addition of HPMC decreased the percentage of drug released. PVP with Poloxamer failed in optimization, while PVA did not show to have any effect on phase transition but increased drug release efficiency [68].

Because of its non-toxicity, biodegradability, biocompatibility, stability and bioadhesive properties, chitosan and related derivatives have been explored as a thermosensitive hydrogel system for biomedical applications in injectable products, whose technological and biological requirements are quite similar to those of ophthalmic formulations [51]. Chitosan is often included in ophthalmic formulations because of its biocompatibility and biodegradability, permeation enhancing and corneal wound healing effects, antimicrobial and antifungal actions and strong mucoadhesive nature. Associating chitosan with stimuli-responsive polymers increases the mechanical strength of the formulation, resulting in better compliance and an increased therapeutic effect [69].

Chung and colleagues [70] proposed either Pluronic-chitosan or PNIPAM-chitosan [71] copolymers as matrices of injectable and thermoreversible hydrogels. A thermosensitive hydrogel system based on chitosan and PVA was also reported [72] and recently, a new chitosan-dibasic orthophosphate hydrogel with thermoreversible gelation was developed by Ta and colleagues [73].

Finally, Hu and colleagues [74] recently proposed a novel hydrogel composed of thermoresponsive block copolymers of methoxy-PEG-PLGA cross-linked with 2,2-bis (2-oxazoline), which had a sol-gel behavior phase transition. Sustained release of bevacizumab was obtained in vitro up to 30 months without burst effect, and after 1 month of intravitreal injection it had no cytotoxicity in vivo on retina cells. Therefore, also thanks to its intraocular characteristics of biodegradability and bioactivity, it appeared to be a promising intravitreal injection carrier for bevacizumab delivery.

### 2.2. Thermosensitive Composites

NP and microparticles (MP) are colloidal systems that, in addition to the advantages similar to those of liposomes, provide flexibility in routes of administration, are tunable in size and have a surface whose functionality can be made responsive to external stimuli [75,76]. MP or NP suspended in a gel matrix (composite systems), generally made of biodegradable polymers, are emerging as a versatile platform in ocular drug delivery applications and may provide reduced side effects, lowering the frequency of intraocular injections [77,78,79].

Tan and colleagues [80] recently developed an interesting thermosensitive in situ gelling formulation, consisting of nanostructured lipid carriers (NLC) incorporated in a hydrogel of hydroxypropyltrimethyl ammonium chloride chitosan and β-glycerophosphate, to form a NLC-loaded hydrogel carrier for ocular sustained release. In in vitro release studies, the resulting dexamethasone-loaded formulation showed a sustained drug release, as 88.65% of the total drug was released from the nanocomposite within 3 days, while almost the same amount was released from simple NLC in 48 h. Therefore, dexamethasone-based NLC gel seemed to be a promising drug delivery system.

Fedorchak et al. [81] developed and tested a combined thermoresponsive hydrogel/microsphere eye drop that was administered like a traditional eye drop, but was able to form a pliable, non-degradable depot after exposure to body temperature in the conjunctival *cul de sac*. Fabiano and co-workers [82] developed a nanocomposite system composed of 5-fluorouracil (5-FU)-loaded NP dispersed in thermosensitive hydrogel made of chitosan and its quaternary ammonium conjugates.

These formulations and that not containing NP were instilled in rabbits, and the 5-FU transcorneal penetration was measured by analyzing the aqueous humor. Both systems increased the area under the curve 3.5 times compared with the phosphate buffer 5-FU solution, chosen as reference. The results underlined the ability of hydrogels to control drug release to a zero order and that of NP to be internalized by cornea cells. Therefore, this nanocomposite formulation can be assessed as an interesting ophthalmic sustained-release DDS. Ammar and colleagues [83] observed the improvement of ocular bioavailability, onset and duration of action of dorzolamide hydrochloride when formulated in an in situ gel nanoemulsion-based delivery system for precorneal application. Poloxamer 407, with its thermoreversible gelation and surface active properties, was utilized to formulate this gel-forming system, and an improvement of ocular drug bioavailability was demonstrated.

While the thermoresponsive composites described above were designed for precorneal treatments, a number of formulations based on MP or NP incorporated into thermosensitive gels are described in the literature as useful injectable systems for cell and therapeutics intraocular delivery (Figure 1).

Gao et al. [84] designed and developed a drug-delivery system containing a combination of poly(d,l-lactide-*co*-glycolide) (PLGA) MP and alginate hydrogel for the sustained release of retinoids to be used in the treatment of retinal blinding diseases. Increased electroretinogram responses were observed after subcutaneous injection of the resulting system in mice, offering a potential alternative to maintain and even to restore vision in humans with certain forms of hereditary blindness.

Hirani and colleagues [85] developed a DDS composed of triamcinolone acetonide-encapsulated PEGylated PLGA NP incorporated into PLGA-PEG-PLGA thermoreversible gel: it resulted in non-cytotoxicity to adult retinal pigment epithelial cell line-19 (ARPE-19) cells and was effective in reducing in vitro VEGF expression when compared to pure triamcinolone.

In another study [86], poly(lactic-*co*-glycolic acid) (PLGA) microspheres obtained by a modified solvent evaporation from the double-emulsion technique were suspended in an injectable poly(*N*-isopropylacrylamide)-based thermo-responsive hydrogel. The developed microsphere-hydrogel DDS resulted in an advantageous ocular DDS for anti-VEGF agents (ranibizumab or aflibercept) with the potential of releasing both the bioactive molecules for 196 days.

Biodegradable doxorubicin (DOX)-loaded poly(d,l-lactide-*co*-glycolide)-poly(ethylene glycol)-folate (PLGA-PEG-FOL) micelles (DOXM) were prepared and suspended in PLGA-PEG-PLGA thermosensitive gel. DOXM exhibited higher cellular uptake than pure DOX in overexpressing folate receptor Y-79 cells, proving their ability to act as a promising targeted delivery system for anticancer agents to retinoblastoma cells following intravitreal administration [87].

Famili et al. [88] developed drug-loaded polymeric micelles dispersed in a thermosensitive gel as an injectable ocular delivery system for triamcinolone acetonide (TA), a very poorly soluble corticosteroid. As a matrix gel they used poly(urethaneurea) backbone, partially grafted with the temperature-sensitive poly(*N*-isopropylacrylamide) (PNIPAAm), whilst micelles were composed of tri-block copolymer, including a high molecular weight polyurethane capped by poly-ethyleneglycol (PEG). To obtain TA-loaded micelles, two methods were employed: a traditional emulsification-sonication procedure and an extrusion method. In vitro tests proved that the combined system significantly improved TA kinetics, sustaining the release for approximately one year. Clinically, this system represents a significant improvement over the current intravitreal injections of corticosteroid suspensions because of its biocompatibility, degradability, long-term TA release profile and ease of administration—indeed, it can be injected through down to 32-gauge needles, well tolerated by patients.

More recently, Agrahari et al. designed composite nanosystems based on biodegradable pentablock copolymers, such as polylactic acid, polyglycolic acid, polycaprolactone and PEG to obtain sustained-release formulations suitable to deliver macromolecules to the posterior eye. In a previous study, pentablock copolymers (PB-1 and PB-2) were successfully synthesized by ring-opening bulk copolymerization of ɛ-caprolactone; then, IgG-Fab was encapsulated into NP of the pentablock copolymer (PB-1 NPs) using the water-in oil-in water double emulsion solvent evaporation method. Finally, IgG-Fab-loaded PB-1 NP were suspended in the thermosensitive PB-2 gelling copolymer. The resulting composite nanoformulation exhibited a sustained delivery, a negligible burst release and a near zero-order release for 80 days in vitro. To assess cell viability and biocompatibility, the same formulation was tested on ocular cell lines and they resulted in being safe for clinical applications [78]. In a second research work, the same authors used different molecular weight polymers (PEG, polylactic acid and polycaprolactone) to prepare NP that were then dispersed in a thermosensitive gel obtained by sequential ring opening bulk polymerization. To evaluate the effect of molecular weights on release profiles, they used three macromolecules (lysozime ~14.5 kDa, IgG-Fab ~50 kDa and IgG ~150 kDa). Results showed a minimal burst release and a sustained drug release from the composite nanosystem; the safety of pentablock copolymers were assessed by in vitro biocompatibility studies, showing the possibility to use these innovative composite nanosystems to deliver macromolecules to the eye for a prolonged time [89].

A summary of all these recently proposed thermosensitive nanocomposites is reported in Table 2, together with the thermosensitive in situ forming hydrogels described above.

### 2.3. Thermosensitive Devices

In addition to thermosensitive formulations, the literature reports the development of microneedles as alternative DDS to topical, systemic or intraocular administration for the treatment of posterior segment diseases. Jiang et al. [90] demonstrated that the insertion of coated microneedles via intrascleral and intracorneal routes could be a useful ocular DDS. An interesting method that combines microneedle technology with the advantages of thermosensitive in situ gelling is described by Thakur and co-workers [91]—they tested a minimally invasive hollow microneedle ex vivo on whole rabbit eyeballs to release in situ self-assembling thermoresponsive poloxamer-based implants into the scleral tissue, which were able to give a sustained drug delivery. The release of fluorescein sodium from these intrasclerally-injected implants was studied, and the implant localization and scleral pore-closure were examined by optical coherence tomography. Sustained release of the model molecule was obtained, varying with the needle heights and the region of intrascleral injection.

Tunc et al. [92] used Parylene C (poly(monochloro-p-xylylene)) and poly(dimethyl siloxane) (PDMS) coated with PNIPAM as implant materials by following pars plana vitrectomy. Three implants of PNIPAM-coated parylene and three of PDMS were inserted over the retina in six rabbits. In vivo results evidenced that PNIPAM-coated implants are able to adhere to the retina without significant ocular toxicity in the short term and can prevent retinal detachment. Thermosensitive hydrophobic acrylic materials such as SmartPLUG, composed of biocompatible and thermosensitive poly(stearylmethacrilate) with methylmethacrylate, have been in use since 2002. These polymers are combined to form materials with glass transition temperature (Tg) or melting temperature (Tm) below human body temperature (37 °C). After administration into the eye, they expand with the subsequent fixation to the patient’s punctum or canaliculum. Recently, punctal plugs (PPs), originally introduced as lacrimal occlusive devices for the management of dry eye disease, have been proposed for ocular DDS [93]. These systems, loaded with poor ocular bioavailability drugs, can be useful to obtain a desired release rate with a significant increase in bioavailability [94]. Moxifloxacin (MOX)-loaded PPs [95] were developed (Ocular Therapeutix, Bedford, MA, USA) to extend the drug release in the bacterial treatment of conjunctivitis.

Hyun Jung Jung and Anuj Chauhan [96] designed temperature-sensitive contact lenses to induce ophthalmic drug delivery. In particular, they prepared timolol-loaded NP and dispersed them in polymer gels. The particle-loaded p-HEMA gels were synthesized by free radical polymerization of the HEMA monomer mixed with the timolol-loaded NP. In vitro release experiments evidenced that NP loaded gels can be used as contact lenses that are able to release the drug upon insertion in the eye.

## 3. In Vitro and In Silico Models to Test Intraocular DDS

Currently there is a lack of convenient platforms for the treatment of posterior eye diseases, as several aspects of ocular drug delivery, such as distribution, clearance and overcoming barriers by drugs, are not completely understood. Meanwhile, corneal and conjunctival epithelia play an important role for topically administered ophthalmic drugs in the development of alternative therapy for the back of the eye. It is important to know the crucial role of the retina, the thin light-sensitive membrane that lines the inner surface of the eye and that, together with the retinal pigment epithelium (RPE), constitutes the blood retinal barrier (BRB) [97]. Intravitreally administered drugs distribute in the vitreous and to the surrounding ocular tissues. Their elimination from the vitreous cavity occurs via blood-ocular barriers and aqueous humor outflow.

In parallel, the knowledge of pharmacokinetic (PK) profiles of intraocular administered drugs is essential to determine the minimum dosing frequency to achieve the maximum therapeutic concentration. Several researchers analyzed the PK parameters of intravitreal-injected drugs in rabbit eyes with the employment of animal tests [98]. Overall, traditional PK studies of eye dissection yielded important information about the mechanisms, even if these approaches are affected by some limitations. The employment of magnetic resonance imaging (MRI) was useful in assessing drug penetration through ocular barriers, the location of ocular and periocular depots, the release kinetics from ocular implants and the clearance process kinetics [99].

Moreover, the use of in vivo models makes the results difficult to evaluate, to compare and to correlate, especially whenever eye dissection is required [100]. The challenge is to design alternative experimentally controlled models that give more reproducible data, reducing animal use, and with consequent cost reductions and ethical advantages.

The current trend is to develop in vitro models able to evaluate PK parameters of drugs considering their physico-chemical properties and by mimicking anatomical and physiological factors. Such models are being increasingly employed in the preclinical stage during drug discovery and development; they can also be used to establish in vitro–in vivo correlations and in quality control procedures. A typical approach is to combine in vitro data with in silico results to build a model whose performance has to be tested in animal experiments [101].

As an example, the USP4 dissolution apparatus has often been used to test the dissolution of poorly soluble drugs and sustained-release tablets [102,103]; meanwhile, in vitro methods based on freshly explanted animal corneas/eye bulbs mounted on diffusion cells are generally employed to evaluate transcorneal and transcleral drug permeation [104,105,106]. Even though these model systems are very useful for initial screening, they have limitations in that they consider neither the human eye compartmentalization nor the mass transfer process induced in the vitreous humor by the outer aqueous flow. From this perspective, Liu and Wang developed an ex vivo method, which allows for the evaluating the precorneal residence time of ophthalmic formulations. This method consisted in a freshly excised rat cornea placed on a chamber perfused with normal saline solution through two precision pumps controlling the in/out flow rates [107].

To simulate the physiological condition, in addition to the perfusion, an efficient model might also mimic the vitreous motion induced by saccadic eye movements. In a recent work, Repetto and colleagues [108] presented a model consisting in a spherical chamber able to rotate at a scheduled time. Based on this apparatus, Bonfiglio et al. [109] developed a magnified scale model of the vitreous chamber suitably modified to evaluate the aspects of drug distribution in the vitreous.

A test method simulating the in vivo situation, called the Vitreous Model, was developed by Loch and colleagues to observe the release behavior of ophthalmic dosage forms, such as intravitreal injections or implants, and the consequential drug distribution in the vitreous body [110]. The same group developed a multi-layer diffusion cell with the purpose of simulating some of the critical parameters that influence drug release and distribution from dosage forms injected in the subconjunctival or intrascleral region [111].

Greater insight is provided by the work of Awwad et al. [112]—these authors proposed an in vitro eye model adapted to human size, consisting in two compartments assembled in such a way as to reproduce the aqueous flow and the mass transfer through the anterior route. This model, named PK-Eye, was employed to assess the PK parameters (clearance, residence time, release profiles, etc.) of therapeutic proteins, such as ranibizumab, bevacizumab and triamcinolone acetonide suspension, and resulted in a promising preclinical tool to study novel long-lasting therapeutics targeted to the posterior segment.

Recently, this model has been refined by our research group [113], which designed and developed a three-compartment ocular flow cell. Similarly to the PK-Eye model cited above, this new model can be filled with simulated vitreous (agar and hyaluronic acid-based); in addition, it contains a semipermeable disk between the central and the posterior section that can act as a support for retinal cells (Figure 2).

In the literature, several papers report the transport behavior of intravitreally administered small and large molecules; results are often obtained by in vivo and ex vivo experiments performed for a few hours. In the case of intravitreally injected protein formulations, it is helpful to follow the stability as a function of time. Accordingly, Patel and colleagues [114], to predict the stability of three intravitreal monoclonal antibodies, developed ex vivo static, semi-dynamic and dynamic models called ExVit.

With the ExVit static model, it was observed that a significant precipitation and aggregation of proteins, probably due to pH level change, occurred in the vitreous humor (VH) after isolation. The semi-dynamic model consisting in two compartments, the VH- and buffer-compartment, effectively stabilized the pH level and facilitated the migration of degradation products. However, the semi-dynamic model did not completely overcome the limitations related to evaluation of long-term protein stability. Therefore, the same researchers designed a dynamic model comprising three diffusion controlling barriers (two membranes and a gel-matrix) to modulate the diffusion rate of macromolecules. More recently, the same group proposed an ex vivo intravitreal horizontal stability model employed to assess the long-term stability of a bi-specific monoclonal antibody (mAb) named ExVit-HS. It consisted of a two-compartment dynamic model (VH- and buffer-compartment) separated by a diffusion controlling membrane (MWCO of 50 kDa) [115]. The vitreous-compartment was filled with VH isolated from porcine eyes with an incision placed near the conjunctiva. The results suggested that the ExVit-HS model can be considered a valuable tool for evaluating long term stability of protein drugs and of other therapeutic molecules that are intravitreally injected.

As previously reported, the fate of the drugs following systemic injection or IVI is influenced by the presence of BRB and RPE permeability. However, the information about this topic is currently controversial. Leena Pitkänen et al. [116] provided permeability values of RPE-choroid as a function of molecule size and lipophilicity. They employed an in vitro diffusion apparatus consisting of a vertical diffusion chamber in which a bovine RPE-choroid was blocked. The permeation data were determined both inward (choroid-to-retina) and outward (retina-to-choroid). Previously, Steuer and colleagues [117] developed a protocol to isolate the porcine BRB in a rapid and gentle way and explain how to immobilize the intact tissue in a two-chamber polycarbonate device. This RPE model showed a large permeability dynamic range, proving to be a valuable tool for research of BRB drug penetration.

Over recent years, some cell culture models mimicking ocular barriers have been proposed and are considered as a useful alternative to in vivo toxicity tests to investigate pathological conditions and the toxicological screening of compounds [118]. However, these models are affected by several intrinsic restrictions, mainly because they are formed by cell monolayers grown on a two-dimensional (2D) culture scaffold, which does not take into account the behavior of cells in the three-dimensional (3D) curved native ocular tissue. Therefore, ophthalmic research is strongly interested in developing 3D ocular in vitro models for toxicity testing, safety screening and evaluating long-term drug effects [119,120]. 3D models are more suitable to create a cell-based platform whose responses will be as representative as those obtained in in vivo conditions.

Recently, Postnikoff et al. [121], in the attempt to reproduce the curved cell growth conditions, cultured human papillomavirus-immortalized cells on a curved Millicell-HA membrane. They obtained a stratified, curved epithelial model and employed it to study the biocompatibility of benzalkonium chloride, a preservative widely employed in commercial eye drops. The results underlined the suitability of this model for biocompatibility experiments.

Nowadays, most of the in vitro ocular models described in this review and summarized in Table 3 are in the pipeline and need further investigation.

Computational modeling can also provide information about drug distribution within the vitreous of animals and humans after injection or release by implants [122,123,124]. However, these mathematical models are limited, as they neither consider nor discuss the stability and interactions of proteins/excipients with the vitreous humor components.

Aapo Tervonen et al. [125] developed a computational model of the physical barrier function of the outer BRB, reflecting the corneal model of Edward and Prausnitz [126] and aiming to relate the properties of the molecule, such as the lipophilicity and radius, to the permeability of the material and to the tissue diffusion pathways. They also introduced a tight junction model structure for the epithelial model.

## 4. Conclusions

Pharmacologists and drug delivery scientists consider ocular drug delivery as a major challenge, owing to the particular anatomy and physiology of the eye and to the presence of various static and dynamic barriers. Topical drug delivery is used generally to treat anterior segment disease, as most of the instilled drop is unable to reach the posterior segment of the eye, meanwhile intraocular injections, mainly IVIs, are considered a gold standard for the treatment of posterior segment diseases. Aside from this, several studies have reported the efficacy of periocular and intravitreal injections, as well as those of intravitreal implants; however, high administration costs hinder their effective application. A number of potential serious risks are also involved, such as retinal detachment, increased intraocular pressure endophthalmitis and cataractogenesis.

Recently, a variety of intraocular controlled-release DDS has been developed with the aim to deliver drugs at a stable concentration over an extended period in patients with conditions affecting the posterior segment, including uveitis, macular edema and AMD. The understanding of such drugs’ PK and the comprehension of the physiological barriers of ocular tissues is essential to the establishment of a valid strategy to reduce side effects of intraocular injections.

The main content of this review was the analysis of the literature, with a special focus on the recently developed thermosensitive systems (hydrogels, composites and devices) proposed as strategies to increase the level of safety and efficacy and to reduce the side effects of ophthalmic treatments. This review also summarized preclinical in vitro models employed to evaluate the performance of innovative intraocular DDS.

## Figures and Tables

**Figure 1 nanomaterials-09-00884-f001:**
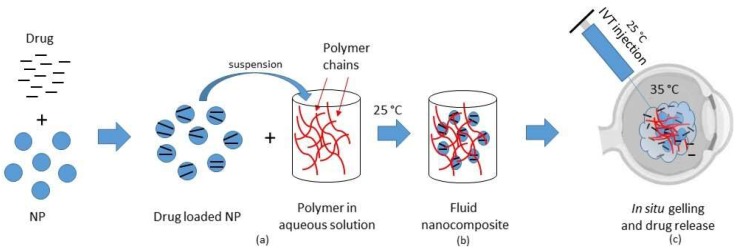
Possible scheme of preparation and intravitreal injection of a thermosensitive nanocomposite system: (**a**) Drug loaded NP are suspended in a polymeric solution at 25 °C. (**b**) Polymeric solution-suspended NP form a fluid injectable nanocomposite system at 25 °C. (**c**) The fluid nanocomposite system is injected in the vitreous at 25 °C; the temperature-induced phase transition occurs at 35 °C, resulting in in situ gel formation and sustained drug release.

**Figure 2 nanomaterials-09-00884-f002:**
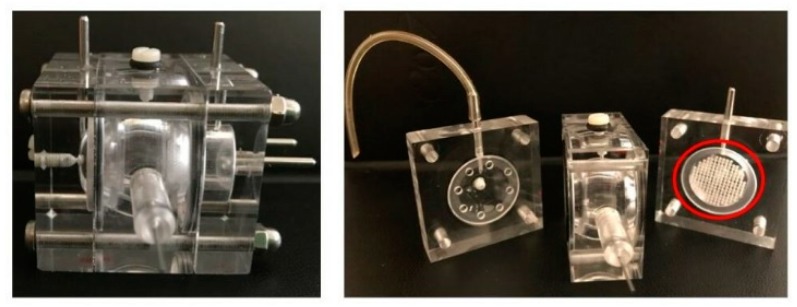
Ocular three-compartments flow cell developed in our laboratory with the semipermeable disk support for retinal cells (red circled).

**Table 1 nanomaterials-09-00884-t001:** Types of thermosensitive drug delivery systems (DDS) for ocular administrations.

Thermosensitive DDS	Mechanism
In situ hydrogels	Formulation is liquid at room temperature (20–25 °C). The contact with body temperature (35–37 °C) leads to the formation of a hydrogel network
Composite systems	Microparticles (MP) or nanoparticles (NP) suspended in a thermogelling matrix generally made of biodegradable thermosensitive polymers
Devices	Polymeric materials (microneedle, punctal plug, contact lens) that, once inserted in the eye, release the loaded drug by change in temperature (from 25 to 35 °C)

**Table 2 nanomaterials-09-00884-t002:** A summary of the thermosensitive systems (hydrogels and composites) developed for ocular drug delivery.

Drug Delivery Systems (DDS)	Material/Aim	Molecule Delivered	Observation	References
In situ hydrogel	Poly(*N*-isopropylacrylamide) (PNIPAM) crosslinked with poly(ethylene glycol) diacrylate (PEG-DA) as an intravitreal injectable vehicle.	Proteins, Ig, Bevacizumab, Ranibizumab	Release profiles function of the cross-link density. Release sustained for approximately 3 weeks.	[54]
In situ hydrogel	Copolymers of *N*-isopropylacrylamide (NIPAM), acrylic acid *N*-hydroxysuccinimide (NAS) and varying concentrations of acryloyloxy dimethyl-c-butyrolactone (DBA) and acrylic acid (AA) for intravitreal injections.	Dexamethasone	Slow-degrading copolymers (over 130 days of incubation in PBS), injectable from a 30-gauge needle, offering slow drug release.	[56,57]
In situ hydrogel	Copolymers of NIPAM and acrylic acid *N*-hydroxysuccinimide (NAS) conjugated with amine-functionalized hyaluronic acid (HA) as a biomaterial scaffolds for bolus injection into the sub-retinal space.	Cells	Temperature-induced scaffold formation for transplanted cells entrapment; optimal compatibility with retinal pigment epithelial.	[58]
In situ hydrogel	PEG-poly-serinol hexamethylene (ESHU) as an intraocular drug-delivery vehicle for age-related macular degeneration (AMD).	Bevacizumab	The release of bevacizumab was sustained up to 17 weeks and its concentration was maintained averaging 4.7 times higher than that in eyes receiving bevacizumab bolus injections. Biodegradable.	[59,60]
In situ hydrogel	Poly(lactic-*co*-glycolic acid) (PLGA)-PEG-PLGA triblock as an intravitreal injectable hydrogel for sustained drug release.	Bevacizumab	Hydrogel immediately formed after intravitreal injection; in vitro sustained drug release over a period of up to 14 days.	[63]
In situ hydrogel	PLGA-PEG-PLGA triblock as an intravitreal injectable hydrogel for sustained drug release.	Dexamethasone	Drug ocular retention time was prolonged from several hours to more than 1 week after a single intravitreal injection; excellent biocompatibility.	[64]
In situ hydrogel	PLA-PCL-PEG pentablock-based injectable hydrogel for the treatment of posterior segment neovascular diseases.	IgG	Significantly longer sustained release of IgG was provided by pentablock (more than 20 days) respect to triblock copolymers.	[65]
In situ hydrogel	Poloxamer 407 in combination with other polymers to form a topical in situ gel.	Ciprofloxacin	Improved antimicrobial effect in vitro compared to the market eye drops. Eight hour sustained release of ciprofloxacin.	[68]
In situ hydrogel	mPEG-PLGA cross-linked with 2,2-bis (2-oxazoline) aqueous solution as an intravitreal injection carrier.	Bevacizumab	Thermoresponsive, controlled drug release. Intraocular biocompatibility biodegradability and bioactivity of loaded drug.	[74]
Nanocarriers inin situ gel	Nanostructured lipid carriers (NLC) in thermoreversible HACC/GP gel for topical ocular delivery.	Dexamethasone	Precorneal sustained release of drug from NLC-HACC/GP gel in vitro.	[80]
In situ gelled nanoemulsion	Poloxamer 407 and Poloxamer 188 in nanoemulsion for topical ocular delivery.	Dorzolamide hydrochloride	Increased precorneal residence time and bioavailability.	[83]
NP in hydrogel	PEG-PLGA NP in PLGA–PEG–PLGA thermoreversible intraocular injectable gel.	Triamcinolone acetonide	Non-toxic and able to reduce vascular endothelial growth factor (VEGF) levels in ARPE-19 cells; sustained release of the drug over 10 days.	[85]
NP in hydrogel	PCL-PLA-PEG-PLA-PCL-based NP in thermosensitive mPEG-PCL-PLA-PCL-PEGm intraocular injectable gel.	IgG-Fab	Minimal burst release with near zero-order release profile from the composite nanoformulation up to 80 days. In vitro cell viability and biocompatibility.	[78]
Micelles in hydrogel	PLGA-MP in poly(*N*-isopropylacrylamide) as an injectable MP-hydrogel drug delivery system.	Ranibizumab and Aflibercept	Controlled and extended intraocular release for approximately 200 days.	[86]
Micelles in hydrogel	Poly(d,l-lactide-*co*-glycolide)-poly(ethylene glycol)-folate (PLGA-PEG-FOL) micelles in PLGA-PEG-PLGA thermoreversible gel for intravitreal administration.	Doxorubicin	Sustained drug release for 2 weeks; increased drug uptake in Y-79 cells overexpressing folate receptors.	[87]
Micelles in hydrogel	PEG-PHS-PEG micelles in PNIPAM-PSHU backbone as an injectable ocular DDS.	Triamcinolone acetonide	Sustained long-term drug release and reduced burst release.	[88]

**Table 3 nanomaterials-09-00884-t003:** In vitro ocular pharmacokinetic models.

Model	Molecules Delivered	Outcomes	References
Dissolution apparatus (USP4) with a 1.5 mL/min flow (8–19 mL chamber)	Model drugs loaded in ocular implant	Adequate sensitivity; correlation with in vivo conditions	[102]
Spherical cavity, magnified with respect to the real geometry, carved within a Perspex cylinder and able to rotate	Blank glycerol solution	First attempt to measure the flow field induced by saccadic eye movements on a model of the vitreous chamber	[108]
Plexiglas cylinder with an internal spherical cavity, magnified with respect to the real geometry, able to rotate	Model drug particles	Three-dimensional (3D) understanding of the saccade-induced steady component of the vitreous flow	[109]
Spherical glass corpus filled with a polyacrylamide gel placed on an altered orbital shaker	Fluorescein sodium solution	Good accordance of the results with the porcine vitreous humor	[110]
Multi-layer diffusion cell composed of three layers placed on top of each other, perfused with buffer using a multi-channel pump	Fluorescein sodium solution	Opportunity to simulate the choroidal and conjunctival blood flow in a simplified setup	[111]
Two-compartment in vitro eye flow model (pharmacokinetic (PK)-Eye)	Dye (Coomassie Brilliant Blue)	To mime the intraocular aqueous outflow for vitreous clearance times estimation for proteins and poorly soluble drugs (injectable suspensions or implants)	[112]
Ex-vivo intravitreal horizontal stability model (ExVit-HS)	Bi-specific monoclonal antibody (mAb)	Valuable tool to evaluate protein and other drugs stability after IVT injection	[115]
Isolated bovine retinal pigment epithelium (RPE)-choroid sealed in a vertical diffusion chamber	FITC-dextran	Useful for pharmacokinetic simulation, particulalrly to evaluate the the retinal entry of drugs after transscleral and systemic delivery	[116]
Reconstructed corneal epithelium in the shape of the regular human cornea	Benzalkonium chloride	Suitable for biocompatibility experiments	[121]

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
