# Peer review of "Ocular Drug Delivery: A Special Focus on the Thermosensitive Approach"

_nanomaterials, 2019, doi:10.3390/nano9060884_

Reviewer 1 Report

Overall the paper covers interesting concepts, although some important changes to focus and structure are recommended.

Abstract – at this point the reviewer was very confused about the contents of the article i.e. the route of administration that was going to be discussed. Just writing “thermoresponsive” for “ocular tissue” is not bound to attract readers unless the abstract states which routes are being considered, or states that ALL routes are being considered.

Lines 33-123 – it is recommended that the authors revise the entire introduction section and list the different routes in a more structured manner. Periocular/trans-scleral drug delivery is also worth mentioning here. The writing here should also be more focused; at the moment, they keep jumping between concepts throughout the introduction which makes for very confusing reading.

Line 66 – the authors mention non-invasive strategies but then go on to discuss implants which require intraocular implants which are definitely invasive. Please revise this section.

Line 80 – repetition with line 42; at this point the reviewer was not expecting any additional comments about the intravitreal. Authors are once again requested to organise the introduction to this paper in a more structured manner.

Lines 93 and 109 – the authors are advised against starting sentences with “anyway”.

Line 121 – revise use of the word “former”; it does not appear to be appropriate here.

Line 194 – please include model when stating “in vitro” or “in vivo” e.g. what cell line, what animal, any particular disease, what apparatus?

Lines 219-225 – this paragraph does not appear to show any links to ophthalmic drug delivery, and the utilised references also do not make any mention of this in their titles. Can the authors clarify its inclusion and relevance to the investigated administration route?

Lines 226-228 –the reviewer expects a more thorough analysis of the paper mentioned here; why is this promising? Also, while the authors have typically used “x et al.” to introduce studies, a change in style was seen in this sentence. Consistency in style is recommended.

Line 241 – when mentioning these articles, please also endeavour to discuss the duration of action i.e. how long was the release sustained for? Are there any comparisons to systems where release isn’t sustained?

Line 245 – please state researcher names rather than say “other researchers”. Also in line 248, is the “reference” just pure 5-FU or is it 5-FU NPs or is it 5-FU NPs in non-derivative chitosan? Just saying “reference” is confusing. Apart from these points however, the study has been well discussed.

Line 253 – there is some inconsistency regarding the placement of references. Please always place references in the same spot e.g. after mention of author name or at the conclusion of the first sentence discussing a particular study.

Figure 1 – a more descriptive figure legend explaining the in situ gelation phenomenon is recommended. The figure itself shows no real difference between the structure of the nanocomposite hydrogel in the beaker and in the eye (final two images).

Line 261 – why is the author’s first name provided? This is unnecessary.

Line 266 – a well discussed reference. Many other references within this manuscript would benefit with this level of depth.

Table 2 – it feels as though the first column of this table could be better structured. It may also be beneficial to state the intended route. Ophthalmic drug delivery encompasses numerous administration strategies and just because a polymer is useful e.g. on the corneal surface won’t guarantee its benefit when used via the other routes.

Author Response

Please, see the attached file

Reviewer 2 Report

The review ''Ocular drug delivery...'' by Sapino et al. compiles relevant literature data about the chosen topics. The manuscript is well written and can be easily followed. It contains 122 references that cover a wide range in time and specific subjects. Besides the Introduction, the chapter 2 (thermosensitive DDS) deals with important already published results, and it is structured in several sub-chapters, as appropriate. Table 2 is probably the most important summary, as it bring out in a single view all the data found in recent literature. Therefore, as the work cover in brief all the required scientific information and it is well organized, the manuscript can be accepted for publication as it is.

Author Response

Round  2

Reviewer 1 Report

The review has been improved for the most part, one point that I did not mention last time but feels out of place is section three. This section looks solely at tests for intraocular administration while the authors have discussed topical, peri-ocular and intravitreal strategies in the manuscript.

The manuscript itself is reasonably long and in fact I was going to suggest trimming it down where possible (e.g. in the introduction section), but in the case of section 3 this would benefit with at least a paragraph on models available to test DDS on the corneal surface or in a trans-scleral fashion. A manuscript that immediately jumps to mind regarding retention on the corneal surface is "PMID: 19521781". That study looked at rat corneas, and larger porcine corneas have also been used in a similar fashion. I don't recall similar models assessing periocular retention.

Author Response

The review has been improved for the most part, one point that I did not mention last time but feels out of place is section three. This section looks solely at tests for intraocular administration while the authors have discussed topical, peri-ocular and intravitreal strategies in the manuscript.

The manuscript itself is reasonably long and in fact I was going to suggest trimming it down where possible (e.g. in the introduction section), but in the case of section 3 this would benefit with at least a paragraph on models available to test DDS on the corneal surface or in a trans-scleral fashion. A manuscript that immediately jumps to mind regarding retention on the corneal surface is "PMID: 19521781". That study looked at rat corneas, and larger porcine corneas have also been used in a similar fashion. I don't recall similar models assessing periocular retention.

Many thanks for your comment, we improved the section regarding transcorneal and transscleral in vitro permeation studies and four inherent references were cited.